# Inverse Bicontinuous and Discontinuous Phases of Lipids, and Membrane Curvature

**DOI:** 10.3390/cells14100716

**Published:** 2025-05-14

**Authors:** John M. Seddon

**Affiliations:** Chemistry Department, Molecular Sciences Research Hub, Imperial College London, Wood Lane, London W12 0BZ, UK; j.seddon@imperial.ac.uk

**Keywords:** lyotropic phase diagrams, interfacial curvature, non-lamellar phases, cubosomes, lipid nanoparticles

## Abstract

In this review article I briefly describe lipid self-assembly, interfacial curvature, and lyotropic phase diagrams. I then go on to describe how the phase behaviour can be controlled, and the structure of lyotropic phases can be tuned, by various parameters such as temperature, hydrostatic pressure, or the addition of amphiphilic molecules such as fatty acids, diacylglycerols, and cholesterol. I then give a few illustrations of how such structures/phases may play roles in lipid-based biotechnologies, and in biomembrane systems.

## 1. Introduction

This article is based on a plenary lecture given at the European Joint Theory/Experiment Meeting on Membranes (EJTEMM 2024) conference, held in Debrecen, Hungary in June 2024. It is not intended to be a comprehensive review of membrane curvature, which has become a vast field over the last 2–3 decades (see for example [1,2]. I note that protein condensates can remodel membranes, inducing shape changes and even fission events [3]. I will also say little here about curvature elasticity, which has been covered in considerable detail in many other papers and reviews, for example, see [4,5,6,7]. Rather, it is my aim to describe qualitatively some of the underlying features of lipid self-assembly, with a focus on curved, non-lamellar structures and phases. I then go on to give a few illustrations of how such structures/phases may play roles in lipid-based biotechnologies, and in biomembrane systems themselves [8].

Lyotropic liquid crystals of 1-, 2-, or 3-dimensional periodicity spontaneously assemble when lipids are mixed with aqueous solvent under various conditions of temperature, pressure, and hydration. The most relevant non-lamellar phases from a biological perspective are the inverse hexagonal H_II_ phase and the inverse cubic phases [9,10,11,12]. In this article, I will say little about the H_II_ phase, although I note in passing that the effect of chain packing stress within this phase has recently been discussed in detail [13]. Cubic phases are observed in various biological membrane systems, and are also closely related both geometrically and topologically to membrane fusion and the formation of membrane channels [14]. There are two quite distinct types of inverse cubic phase: bicontinuous ones based on underlying periodic minimal surfaces, and discontinuous ones based on simple or more complex packings of discreet inverse micelles.

In this article I will briefly review lipid self-assembly, interfacial curvature, and lyotropic phase diagrams. I will then go on to describe how the phase behaviour can be controlled, and the structure of lyotropic phases can be tuned, by various parameters such as temperature, hydrostatic pressure, or the addition of amphiphilic molecules such as fatty acids, diacylglycerols, and cholesterol. I will illustrate these effects using examples primarily from work done in the Membrane Biophysics group at Imperial College London over the last few years. There is now a very large literature on non-lamellar phases of lipids, and I do not here attempt to provide a comprehensive review of the field.

For potential medical applications, bulk lipid phases can be dispersed into lipid nanoparticles (LNPs) of the order of 100–200 nm in diameter [15,16]. These are named hexosomes when formed from the H_II_ phase, cubosomes when based on inverse bicontinuous cubic phases, and micellosomes when based on discontinuous cubic phases. It is important to consider whether the internal structure has been disrupted or modified upon converting bulk phases into LNPs. Self-assembled LNPs containing ionisable cationic lipids, with inversely curved internal structures, have recently been spectacularly successful in delivering active molecules such as nucleic acids into cells, the first example being the LNP-encapsulated siRNA drug Onpattro (Patsiran), which was approved in 2018 [17]. The outstanding examples of this approach are the mRNA-LNP COVID-19 vaccines, which have saved millions of lives worldwide [18].

## 2. Amphiphiles and Lipid Membrane Self-Assembly

Lipid molecules that undergo self-assembly into bilayer membranes, or other lyotropic liquid-crystalline phases, are invariably amphiphilic, having distinct regions that are hydrophilic in nature, and others that are hydrophobic. Typical examples of this are the phospholipids, where a polar headgroup is linked to one or more hydrocarbon chains, often via a glycerol linker. Examples of the three most abundant classes of phospholipid found in most animal cell plasma membranes are shown in Figure 1. DOPE is dioleoyl (di-C18 *cis*-unsaturated) phosphatidylethanolamine, DPPC is dipalmitoyl (di-C16 saturated) phosphatidylcholine, and the sphingomyelin shown is the palmitoyl version of this phospholipid. The polar/non-polar interface of these molecules is located close to the average positions of the chain carbonyl groups, as shown in Figure 1.

Although superficially these three classes of phospholipid look quite similar to one another in terms of their chemical structures, and each has a zwitterionic headgroup, in fact they exhibit quite dramatically different self-assembly properties in aqueous solution. This can to a significant extent be attributed to the difference in hydrogen-bonding capacities of these three phospholipids. PC and sphingomyelin have the same phosphocholine polar headgroups, whose phosphate groups can act as hydrogen-bond acceptors but not donors. Given that they only differ in their interfacial regions, they have relatively similar lyotropic phase behaviour, both having a very strong tendency to form bilayer structures—either ordered gel or fluid—over a very wide range of thermodynamic conditions (temperature, hydrostatic pressure, hydration, etc.). The polar headgroup of PE, on the other hand, in addition to the phosphate group, also has an ammonium NH_3_^+^ terminal group, which can act as a H-bond donor and form up to three H-bonds, either to water or to neighbouring PE headgroup phosphate groups. This tends to cause the PE headgroup to be less strongly hydrated than PC or sphingomyelin headgroups. In addition, the molecular volume of the PE headgroup is significantly lower. These effects lead to PE molecules tending to have ‘inverse cone’ shapes (see below), favouring the formation of non-bilayer structures and phases.

## 3. Interfacial Curvature and Lyotropic Phase Diagrams

In considering membrane curvature, we need to define both the interfacial mean curvature *H*, and the Gaussian curvature *K*. The former is defined at a given point on an interface as the average of the two principal curvatures *c*_1_ and *c*_2_ at that point, and the latter is defined as the product of *c*_1_ and *c*_2_ (Figure 2). The sign of *H* is arbitrary: we here take it to be positive for curvature of the interface of a lipid monolayer towards the hydrocarbon chains (i.e., as in a normal micelle). Inverse non-lamellar phases will have negative *H*. The sign of *K* characterises the nature of the interface:When *c*_1_ and *c*_2_ have the same sign (both positive or both negative), *K* > 0, and the interface is elliptic (e.g., a spherical lipid vesicle).When either *c*_1_ or *c*_2_ is zero, *K* = 0, and the interface is parabolic (e.g., flat bilayers, or cylinders as in the inverse hexagonal H_II_ phase.)When *c*_1_ and *c*_2_ have opposite signs, *K* < 0, and the interface is hyperbolic (e.g., a saddle-surface), as found transiently in membrane fusion channels, and stably within bicontinuous cubic phases.

The preferred interfacial mean curvature of a lipid monolayer is set by the average ‘shape’ of the lipid molecules making up the monolayer (Figure 3). When the average cross-sectional area in the lipid headgroup region is larger than that in the chain region, as shown on the left, the lipids are ‘cone-shaped’ and tend to pack into monolayers having a positive mean curvature (curvature towards the chain region), whereas when the headgroups are less bulky in cross-section than the chains, as shown on the right, the lipids are ‘inverse cone-shaped’, and tend to pack into monolayers that have a negative or inverse curvature, away from the chains and towards the polar region. When the cross-sections in the polar and chain regions of the lipid are roughly equal, the average molecular shape is cylindrical, favouring flat monolayers (of zero mean curvature). Note that if two identical lipid monolayers have a preferred non-zero value of *H*, if they come together back-to-back to form a bilayer (due to the hydrophobic effect), the bilayer must be flat on average by symmetry (although its mean curvature *H* can undergo thermal fluctuations). Thus, the lipid bilayer will contain an inherent curvature frustration due to the monolayers been held away from their preferred state of curvature.

The average molecular ‘shape’ of a given lipid depends not only on its chemical structure, but also on the thermodynamic conditions (temperature, hydrostatic pressure, hydration, solution pH, etc.). Attempts have been made to determine the lipid ‘packing parameter’ from molecular dynamics simulations [20]. Increasing temperature increases the chain conformational disorder, leading to an increased splay in the hydrocarbon chain region; conversely, increasing hydrostatic pressure tends to have the opposite effect, reducing the splay, since it tends to order the chains. Starting from a flat fluid lipid bilayer (where both monolayers are also forced to be flat), increasing temperature will tend to induce transitions towards nonlamellar phases having increasingly negative (inverse) interfacial mean curvature. These transitions may then generally be reversed by application of raised hydrostatic pressure, reducing the preferred interfacial curvature.

It is conceptually useful to arrange the fluid lipid lyotropic phases according to the average interfacial mean curvature at the polar/nonpolar interface within each phase (Figure 4). This interface corresponds to the position where the lipid polar headgroups are connected to their hydrocarbon chains and is not sharply defined. The fluid lamellar L_a_ phase, having zero average interfacial mean curvature, occupies a central position in this schematic phase sequence. The phases to the right have increasingly positive mean curvature, for example the normal hexagonal H_I_ phase. This region tends to be occupied by single-chain surfactants such as lyso-phospholipids, and will not be further considered here. The phases to the left have increasingly negative mean curvature, for example the inverse hexagonal H_II_ phase. In region *b* are found the inverse bicontinuous phases, and in region *a* are found phases based on 3-D packings of inverse micelles: examples of both of these will be described below.

For some systems the interfacial mean curvature can be tuned by varying the water content, thereby driving phase transitions; for other systems, a more effective parameter is to vary the temperature (or the hydrostatic pressure)—this is particularly the case for inverse lipid phases. We will see later that varying the solution pH can also be a potent controller of interfacial curvature.

## 4. Inverse Bicontinuous Cubic Phases of Lipids

The most commonly found phases in region *b* of the ‘phase diagram’ of Figure 4 are three geometrically closelyt related inverse bicontinuous cubic phases of crystallographic spacegroups Ia3d, Im3m, and Pn3m [21]. These three cubic phases are all based upon underlying periodic minimal surfaces [22], the gyroid (G), primitive (P), and the diamond (D) minimal surface, respectively (Figure 5). In each cubic phase, a continuous fluid lipid bilayer of thickness in the region of 40 Å is ‘draped’ upon a curved periodic minimal surface (which lies at the centre plane of the bilayer), forming ordered sponge-like structures. Each cubic phase contains two interpenetrating networks of water channels, threefold, sixfold, and fourfold connected, respectively. The minimal surfaces have zero mean curvature at all points, and a Gaussian curvature that is everywhere negative, varying from most negative at saddle points, to zero at flat points. It is possible to transform between these three minimal surfaces by carrying out a one-to-one mapping between identical surface patches. In other words, all three underlying minimal surfaces have exactly the same distribution of Gaussian curvature (and identical (zero) mean curvature) but are arranged differently in space; the G minimal surface is the most compact, and the P surface is the most expanded. This means that we expect to see the Ia3d phase at lower hydration and the Im3m phase at highest hydration. This is borne out in practice [23,24]. We show below how we can induce phase transitions between these cubic phases, for example by applying hydrostatic pressure (see Section 9).

In model lipid systems, these inverse bicontinuous cubic phases typically have lattice parameters in the range 100–200 Å, with relatively small water channels in the region of 30 Å. By incorporation of charged phospholipids, we have been able to swell inverse bicontinuous cubic phases to lattice parameters of approx. 500 Å, with water channels of approx. 220 Å diameter, potentially expanding the range of usefulness of such phases for applications such as drug delivery or encapsulation of enzymes [26,27]. Others have also reported similar effects [28]. For example, Figure 6 shows the swelling of a monoolein inverse bicontinuous cubic phase by incorporation of the charged phospholipid dioleoyl phosphatidylserine (DOPS). It is striking that the addition of as little as 1 mol% DOPS switches the D cubic phase (Pn3m) to a more expanded P cubic phase (Im3m).

Interestingly, it is also possible, to a lesser extent, to swell the monoolein cubic phase by the addition of cholesterol (Figure 7). Initially, the Pn3m (D) cubic phase swells, and then in the region of 15–25 mol% cholesterol, undergoes a phase transition to the Im3m (P) cubic phase [26]. A similar result was previously reported by Cherezov and co-workers [29].

The underlying mechanism of the swelling of the monoolein cubic phase is not entirely clear, as cholesterol tends to reduce chain splay (which would favour swelling), but can also act as an ‘inverse phase’ promotor, for example when added to dioleoyl phosphatidylcholine [30].

## 5. Relationship Between Membrane Curvature and Membrane Fusion

Fusion between two flat lipid bilayer membranes involves a change in topology. A fusion channel between two such membranes has a negative Gaussian curvature (saddle curvature) in the region of the pore (Figure 8).

There is an intimate relationship with bicontinuous cubic phases [31], as the latter are essentially 3-D ordered arrangements of fusion pores (see Figure 5). Formation of such a pore will be energetically favoured if the bilayer Gaussian curvature modulus is positive, which can occur when the two lipid monolayers have sufficiently negative spontaneous curvature (tendency for inverse mean curvature). A recent paper has proposed a novel method for determining intrinsic lipid curvatures [32].

## 6. In Cubo Crystallisation of Membrane Proteins

The technique of in cubo (or in meso) crystallisation of membrane proteins (Figure 9) was first demonstrated by Landau and Rosenberg [33]. It was further developed by Caffrey, Cherzov, and others [34] and was successfully employed to crystallize a GPCR (G protein-coupled receptor) complex [35], among various other membrane protein systems.

## 7. Complex Membrane Morphologies in Biological Systems

Deng and co-workers have detected cubic membranes within a range of organelles in cells, for example the mitochondria of the amoeba *C. carolinensis* were studied by electron tomography, and it was found that the cristae exhibit bicontinuous cubic membrane ordering, with a lattice parameter in excess of 1000 Å [37]. This group has discovered a range of other biological systems exhibiting cubic membrane ordering, for example in the mitochondria of the retinal cones of two species of tree shrew [38]. They have even suggested that infection of cells by lipid-enveloped coronaviruses may induce the formation of cubic membranes within the infected cells [39].

In various insects, there are cases of cubic phases that exceed 3000 Å in lattice parameter. For example, in the wing cells of certain butterflies, a complex infolding of the plasma membrane together with the smooth endoplasmic reticulum develops into a highly swollen gyroid cubic membrane morphology, with one of the water networks in contact with the extracellular space. Chitin is polymerized within this network to form a single-gyroid biophotonic crystal (Figure 10), which acts as an optical bandgap material, leading to the irridescent colours of the wings [40,41,42].

This is an example of ‘structural colour’, which is quite common throughout many species of insects, birds, and even certain plants.

Another biological system that exhibits structures that are closely related to periodic minimal surfaces and cubic phases are the endoskeletons of sea urchins and starfish, where structures closely related to the D and P periodic minimal surfaces are observed, with enormous lattice parameters of 30 μm or greater. For example, Figure 11 shows an X-ray tomograph of a region of the endoskeleton of the sea urchin *C. rugosa* [44]. It should be noted that the lattice parameters of these structures are much larger than could be attributed to templating via a bicontinuous cubic phase, and some other soft matter templating mechanism must be involved.

## 8. Inverse Discontinuous Lipid Phases

In region *a* of the ‘phase diagram’ of Figure 4, we expect to find phases that have a more negative inverse interfacial curvature than in the H_II_ phase. To achieve this implies that the inverse cylinders of the H_II_ phase must break up into shorter, or even spherical aggregates, which can then pack onto 3-D lattices [45,46].

We have previously shown that by addition of weakly polar amphiphiles such as diacylglycerols to phospholipids, we can tune the interfacial curvature to be strongly inverse, leading to the formation of a discontinuous cubic phase of spacegroup Fd3m, with a structure proposed to be based upon a complex close packing of two types of quasi-spherical inverse micelles [47]. An example of the characteristic X-ray diffraction pattern from this phase, from a fully hydrated 1:2 mixture of dioleoyl phosphatidylcholine/dioleoyl glycerol (DOPC/DOG), is shown in Figure 12.

The detailed structure of this cubic phase was determined both by X-ray diffraction [49] and by freeze-fracture electron microscopy [50]. The schematic structure (Figure 13) consists of a close-packing of two types of quasi-spherical inverse micelles, with eight larger (red) and 16 smaller ones (blue) per unit cell. The lattice parameter is typically in the region of 150 Å.

This structure has turned out to be the most common lipid phase based upon an ordered packing of inverse micellar aggregates, rather than simpler fcc or bcc packings of spherical inverse micelles. The reason for this appears to be because the degree of chain packing frustration in the latter two structures is too large. In the Fd3m packing, both inverse micelles are close to spherical, minimising the energetically costly variation in chain length required to uniformly fill the hydrophobic region of the phase [51,52]. It is fascinating that nature is able to find a solution to optimise curvature and chain packing by the lipids self-assembling into two types of inverse micelles of different sizes, with a specific number of each micelle per unit cell. This process is facilitated when two lipids having quite different intrinsic curvatures (such as diacyl glycerol and phosphatidylcholine as in the lipid system of Figure 12) are present in the mixture, presumably because the two lipids can partially preferentially locate in one or other of the inverse micelles, according to their preferred curvature. However, there are a few cases, for example a glycolipid/water system, of a single lipid that can adopt the Fd3m phase [53].

A further example of an inverse micellar cubic phase was discovered in a phospholipid/water system in the presence of small amounts of organic solvents such as isooctane, cyclohexane, or limonene [54]. This phase is of spacegroup Fm3m, and its structure consists of an *fcc* packing of identical spherical inverse micelles. Undoubtedly, the role of the organic solvent is to relieve packing frustration within the hydrophobic region of the phase.

Some time ago we discovered a non-cubic inverse micellar lyotropic phase in a hydrated mixture of dioleoyl phosphatidylcholine, dioleoyl glycerol, and cholesterol, between 16 and 52 °C [55]. The characteristic synchrotron small-angle X-ray diffraction pattern is shown in Figure 14 (left).

The diffraction pattern was indexed as 3-D hexagonal, of spacegroup P6_3_/mmc, with lattice parameters *a* = 71.5 Å; *c* = 116.5 Å. The structure was deduced to consist of an *hcp* periodic packing of identical spherical inverse micelles, as shown schematically in Figure 14 (right). This phase is stable in excess aqueous solution over a wide range of temperature and hydrostatic pressure.

This 3-D hexagonal inverse micellar phase is expected to have a greater chain packing frustration than the Fd3m cubic phase, and we proposed that the role of the cholesterol is to relieve the chain packing frustration within the hydrophobic region of this phase, allowing the P6_3_/mmc phase to form. This is plausible, as it is known that the barrier for cholesterol to cross the hydrophobic interior of lipid phases is small, leading, for example, to very fast flip-flop rates (sub-millisecond timescale) across lipid bilayers [56,57].

## 9. Hydrostatic Pressure Effects on Lipid Phase Behaviour

Hydrostatic pressure tends to have the opposite effect to temperature on lipid phase interfacial curvature, since increasing pressure reduces hydrocarbon chain disorder, and hence reduces molecular splay, whereas increasing temperature increases chain disorder and hence increases chain splay. Indeed, in an early study with R. Winter, we demonstrated that an Im3m cubic phase of a 1:2 mixture of dimyristoyl phosphatidylcholine/myristic acid in excess water could be swollen by some 60 Å by application of 1 kbar of hydrostatic pressure [58]. The explanation for this somewhat counterintuitive result is presumably that the reduced chain splay reduces the preferred mean curvature at the polar headgroup region, which can be achieved by the phase sucking in more water from the excess water pool. This offers a way to adjust the diameter of the water channels of the cubic phase isothermally, which could be useful for certain applications. In later work we showed that pressure-jumps could be used to induce transitions between different liquid-crystalline phases, including cubic-cubic phase transitions [59,60,61,62,63,64]. For example, Figure 15 shows time-resolved synchrotron small-angle X-ray diffraction patterns from monoolein, showing an Ia3d (G) cubic to Pn3m (D) cubic phase transition, induced by a pressure-jump from 600 to 240 bar at 59.5 °C [62]. Such measurements permit the kinetics of the phase transition to be determined and can provide evidence for intermediate structures forming during the transition (the transient peak labelled (*a*) was not identified, and the peak labelled (*b*) was from a longer-lived inverse hexagonal phase, although this phase is not expected to be an intermediate structure in a cubic-cubic transition).

Hydrostatic pressure also has striking effects on the structure and stability of the Fd3m cubic phase of a fully hydrated 1:2 mixture of dioleoyl phosphatidylcholine/dioleoyl glycerol [48]. Little or no change is seen up to a pressure of 2 kbar, whereupon there is an abrupt phase transition/separation to coexisting ordered lamellar and H_II_ phases.

## 10. Lipid Nanoparticles: Cubosomes, Hexosomes, and Micellosomes

Bulk lipid liquid-crystalline non-lamellar phases can be dispersed in aqueous solution into lipid nanoparticles (LNPs), of sizes of the order of 100–200 nm, by sonication [15]. Incorporation of a small amount of an amphiphilic triblock copolymer stabilized the lipid particles in solution. A recent review highlights advances in LNP formulations as drug delivery platforms [65]. Lipid nanoparticles based on inverse bicontinuous cubic phases are termed cubosomes (see [66] for a recent review), those based on the inverse hexagonal H_II_ phase are termed hexosomes, and those based on inverse ordered micellar phases are termed micellosomes. A key question is to what extent the internal structure of the bulk phases is preserved within the lipid nanoparticle. This can be assessed by a combination of small-angle X-ray diffraction and cryo-electron microscopy. As shown in the following cryo-EM image of an Im3m cubosome of monololein with 5 wt% of the block copolymer F127 (Figure 16), the internal periodic structure of the cubic phase can be highly preserved in the interior of the cubosome, with characteristic blebs at the surface, where the cubosome is adapting to being in contact with the external aqueous solution.

The structure of cubosomes has been theoretically modelled by Gozdz, using an approach based upon a free energy functional [67]. He finds a host of interesting effects and concludes that the internal structure of small cubosomes can be very rich (Figure 17). It is interesting to note that only one of the two networks of water channels is open to the external solution, whereas the other network is closed off at the surface.

A very detailed study of cubosomes by cryo-electron tomography has shed considerable light on both the internal structure, and how the surface of the cubosome adapts to being in contact with the external aqueous phase [68].

Barriga has shown that cubosomes can be swollen by the incorporation of charged lipids and also by cholesterol, and can show increased incorporation of the lectin PHA-L, a tetrameric protein of 120 kDa, upon swelling [69]. FRET data between Alexa 488-labelled lectin PHA-L and rhodamine-labelled phospholipid in the cubosome were consistent with only one of the swollen water channels being accessible to the external aqueous solution.

Leal and co-workers showed that cubosomes encapsulating siRNA can be taken up by cells and undergo endosomal escape, leading to efficient gene silencing [70,71]. They attributed this to a positive Gaussian curvature modulus of the lipid membrane within the gyroid inverse bicontinuous cubic phase giving the cubosome an inherent fusogenicity. In support of this idea, Tyler and co-workers have recently demonstrated that ‘topologically active’ monoolein cubosomes can drive morphological and topological changes, such as tubulation, budding, and fusion, in giant unilamellar vesicles [72].

There has been considerable interest in the potential involvement of non-lamellar lyotropic phases forming in the gut during digestion of lipid-based foods. A recent review [73] summarises the findings determined by conducting in vitro lipolysis coupled to X-ray diffraction, from various foodstuffs, ranging from milk, soy juice, mayonnaise, and even infant formula. Intriguingly, in all of these cases, the Fd3m inverse micellar cubic phase (as well as various other phases) was observed to form during digestion. The biological significance of this is not well understood at present.

Exposure of Fd3m micellosomes of a soy phosphatidylcholine/dioleoylglycerol mixture to a triacylglycerol lipase, which led to progressive lipolytic degradation of the dioleoylglycerol towards monoglyceride, free fatty acid, and glycerol, led to a progressive change in the interior structure from Fd3m cubic towards less inversely curved phases (H_II_, inverse bicontinuous cubic, L_3_ sponge phase), finally resulting in the formation of multilamellar vesicles [74].

## 11. Microfluidic Platforms for Production of Lipid Nanoparticles

It is possible to produce monodisperse lipid nanoparticles by using microfluidics technologies [75]. For example, Leal and co-workers were able to produce monodisperse cubosomes and then complex the fully formed lipid nanoparticles with gene-silencing RNA [76]. A microfluidic hydrodynamic focussing technology (Figure 18) has been recently developed for the production of cubosomes and hexosomes, whose size is relatively monodisperse and can be controlled by varying the flow rate ratio between the aqueous buffer and ethanolic streams [77]. A similar approach, using a commercial NanoAssemblr microfluidic platform, gave rather equivalent results [78].

## 12. pH-Triggered Changes in Connectivity Within Lipid Nanoparticles

Tyler and co-workers [79] have recently demonstrated that Fd3m micellosomes can be formed in buffer at pH 7.4 by mixtures of monoolein and oleyl alcohol, containing a small amount (3 mol%) of a cationic ionizable lipid (DOBAQ), with a pKa (the pH at which the ionizable group is half-dissociated) of approximately 6. By lowering the pH to below pH 6, the zwitterionic lipid becomes positively charged, triggering a phase transition within the lipid nanoparticle from an internally confined Fd3m structure (micellosome), to a porous inverse hexagonal H_II_ phase (hexosome), favouring release of any encapsulated contents (Figure 19). A combination of small-angle X-ray scattering and cryo-TEM was used to determine the detailed internal structure within the Fd3m micellosomes.

## 13. pH-Dependent Phase Transitions in Cationic Ionizable Lipid/Cholesterol Mixtures

The LNPs used for COVID mRNA vaccines typically contain distearoyl phosphatidylcholine (DSPC), a PEG-lipid, cholesterol, and one of a variety of ionizable amino lipids (for example, ALC-0315 (Pfizer Inc, 66 Hudson Boulevard East, New York, NY 10001-2192, USA/BioNTech SE, An der Goldgrube 12, 55131 Mainz, Germany) or SM-102 (Moderna Inc, 200 Technology Square, Cambridge, MA 02139, USA), which become cationic on lowering the pH below approximately 6–7. The DSPC (helper) and PEG-lipid (stabilizer) reside predominantly in or close to the surface lipid monolayer coating the LNP, while the cholesterol and ionizable lipids are mainly located in the LNP core, the latter being complexed with the encapsulated RNA (Figure 20).

It is believed that the RNA-LNPs taken up into endosomes within cells undergo endosomal escape by pH-induced structural changes as the pH within the endosome is progressively lowered by proton pumps in the endosomal membrane. As the ionizable cationic lipids become increasingly positively charged, some of them interact with negatively charged phospholipids in the endosomal bilayer, leading to localized transformations to non-bilayer structures or pores, allowing the RNA to escape into the cytoplasm of the cell.

Raedler and colleagues have studied the bulk phase behaviour of three cationic ionizable lipids, MC3 (used in Onpattro), KC2, and DD, mixed with cholesterol in 3:1 molar ratios in buffer solutions, as the pH was lowered from 7.3 to 4.5 [80]. It is fascinating that the sequence of phases observed (Figure 21) with lowering the pH (thereby causing the ionizable lipid headgroups to become positively charged), shows first an ordering of inverse micelles onto lattices (3-D hexagonal spacegroup P6_3_/mmc, then face-centred cubic spacegroup Fd3m), followed by a transition to the less inversely curved H_II_ phase by pH 5.5. In the case of the lipid DD, further lowering of the pH below 5.0 induced a transition to the inverse bicontinuous cubic phase Pn3m. This sequence of phases can be compared with the schematic ‘phase diagram’ of Figure 4, which makes it clear that lowering the pH is driving the preferred interfacial mean curvature to progressively less negative values.

These authors have recently extended this work to correlate the buffer specificity of transfection efficiency of LNPs formulated with the cationic ionizable lipids MC3, SM-102 (Moderna), and ALC-0315 (Pfizer/BioNTech) with the pH-dependent structures of bulk phases of ionizable lipid/cholesterol mixtures [81]. Citrate buffer was found to promote the transition to the H_II_ phase to a higher pH and to promote endosomal escape, as indicated by an increased transfection efficiency.

Drummond and colleagues [82] have used X-ray diffraction to study how the internal structure of LNPs formed from the ionizable lipids ALC-0315 (Pfizer/BioNTech) and SM-102 (Moderna) vary with solution pH. They found that ALC-0315 LNPs undergo sequential transitions first to the inverse micellar solution at pH 5.5, then to the H_II_ phase by pH 5, followed by bicontinuous cubic phases Pn3m and Ia3d by pH 3. SM-102 LNPs had a somewhat different phase behaviour, showing an inverse micellar solution phase by pH 6, an H_II_ phase at pH 5.5, and an Ia3d cubic phase at pH 5. On the basis of these results, they proposed a mechanistic scheme (Figure 22) whereby the low pH-induced formation of inverse curved phases in the LNPs promotes destabilization of the endosomal membrane via fusion between it and the LNP, leading to endosomal escape of the LNP contents. This model of endosomal escape is quite similar to that proposed by Leal and co-workers [83].

In summary, this review article has described the self-assembly of phospholipids in aqueous solution into a rich variety of lyotropic liquid-crystalline phases. I have focussed attention on inverse nonlamellar phases, having negative interfacial mean curvature. I have discussed how the phase behaviour can be controlled, and the structures of the phases tuned, either by varying thermodynamic parameters such as temperature, hydrostatic pressure, hydration, or solution pH, or by adding various amphiphilic molecules such as diacylgycerols, cholesterol, or ionizable lipids. I have shown how such nonlamellar phases are intimately related to dynamic processes involving topological transformations in membranes, such as membrane fusion. Furthermore, there is now compelling evidence that nonlamellar phases are utilized by Nature in certain specialized structures in cells, for example in templating the structures which exhibit structural colour in the wing cells of certain butterflies. There is much still to be discovered about the role of such complex self-assembled lipid structures in Nature. Nonlamellar lipid phases, when dispersed into lipid nanoparticles, also very attractive candidates as vehicles for the delivery of drugs and nucleic acids into cells. We can expect to see the highly successful lipid nanoparticle approach taken for COVID vaccines to be extended to many other areas of molecular medicine in the near future.

## Figures and Tables

**Figure 1 cells-14-00716-f001:**
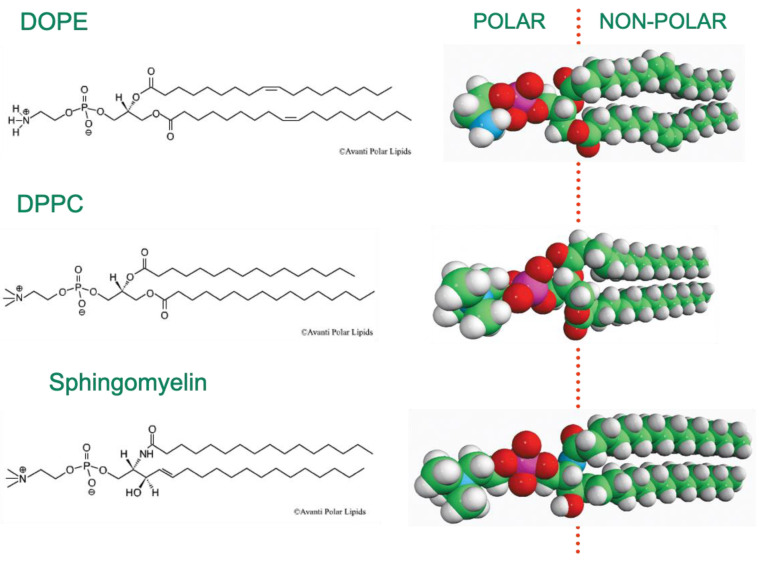
The chemical structures of examples of the three of the most common classes of phospholipid (PE, PC, and sphingomyelin) found in the plasma membranes of animal cells, with the red dotted line on the right indicating the approximate location of the polar/non-polar interface. The chemical structures of the phospholipids shown are reproduced with the kind permission of Avanti Research^TM^.

**Figure 2 cells-14-00716-f002:**
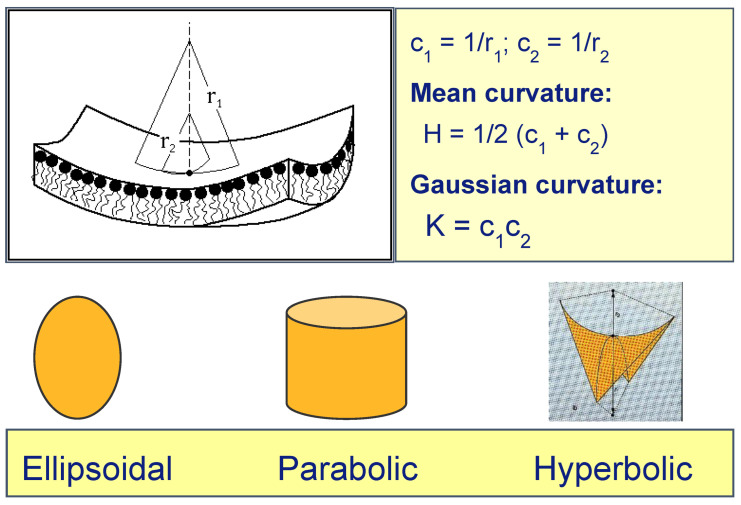
Interfacial curvature at a given point on a lipid monolayer.

**Figure 3 cells-14-00716-f003:**
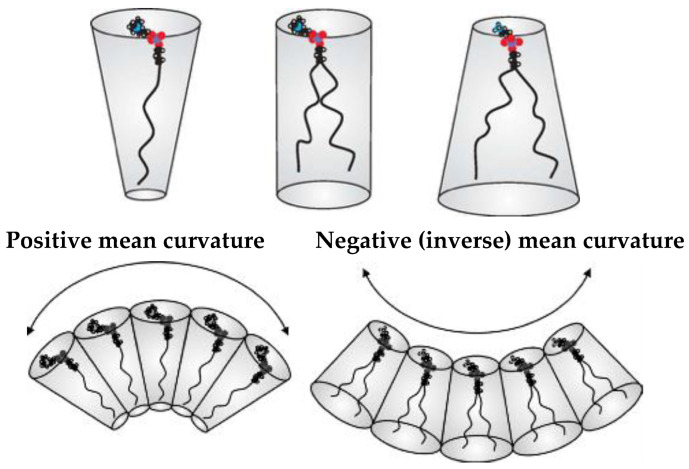
Average lipid ‘shape’ and different signs of interfacial mean curvatures. Adapted from: [19] and used with permission of the Institute of Physics (UK).

**Figure 4 cells-14-00716-f004:**
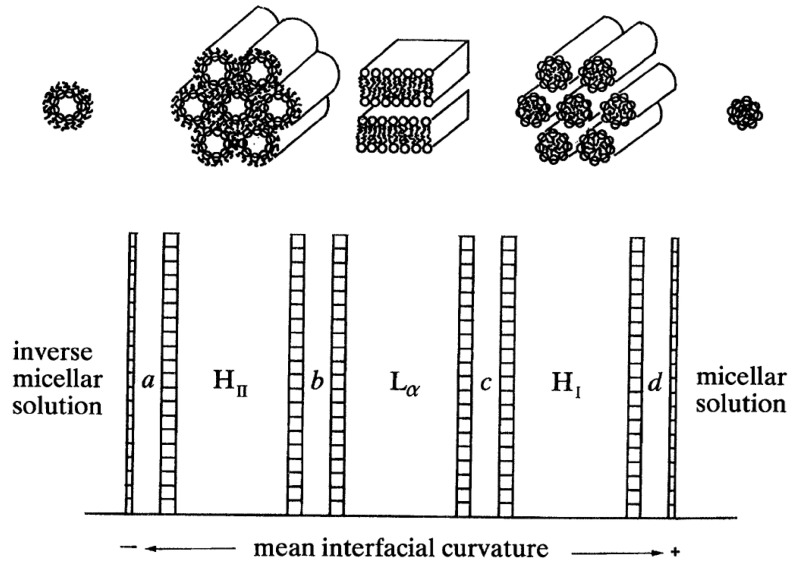
Schematic of the expected sequence of lyotropic phases as a function of the interfacial mean curvature at the polar-nonpolar interface. Taken from [21] and used with permission of the Royal Society (UK).

**Figure 5 cells-14-00716-f005:**
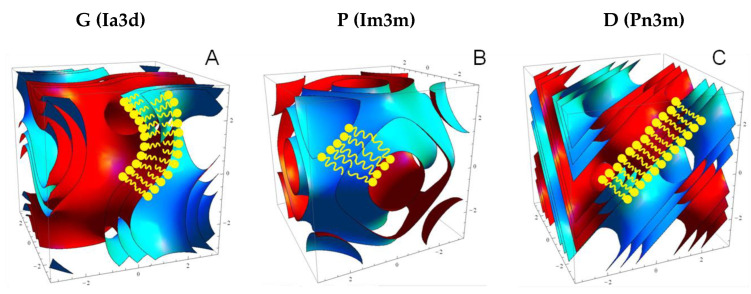
The three geometrically closely related inverse bicontinuous cubic phases of spacegroups (**A**) Ia3d, (**B**) Im3m, and (**C**) Pn3m. Reproduced from [25] with permission from the Royal Society of Chemistry (UK).

**Figure 6 cells-14-00716-f006:**
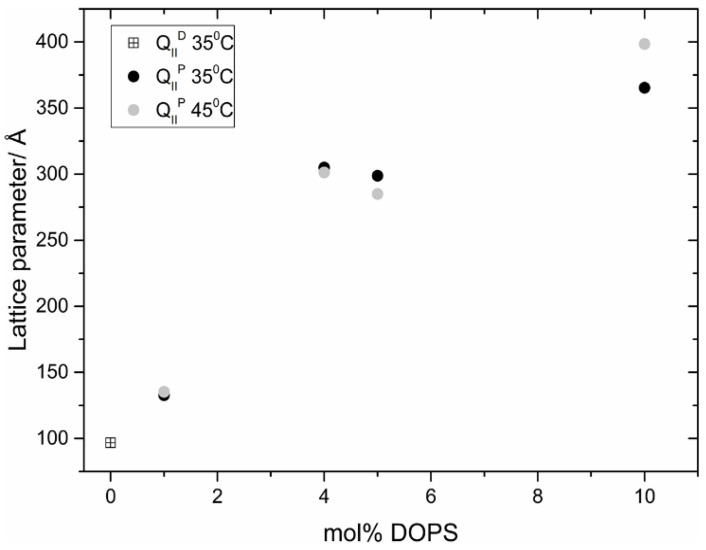
Swelling of a monoolein inverse bicontinuous cubic phase by incorporation of the charged phospholipid dioleoyl phosphatidylserine (DOPS). Reproduced from [26] with permission from the Royal Society of Chemistry (UK).

**Figure 7 cells-14-00716-f007:**
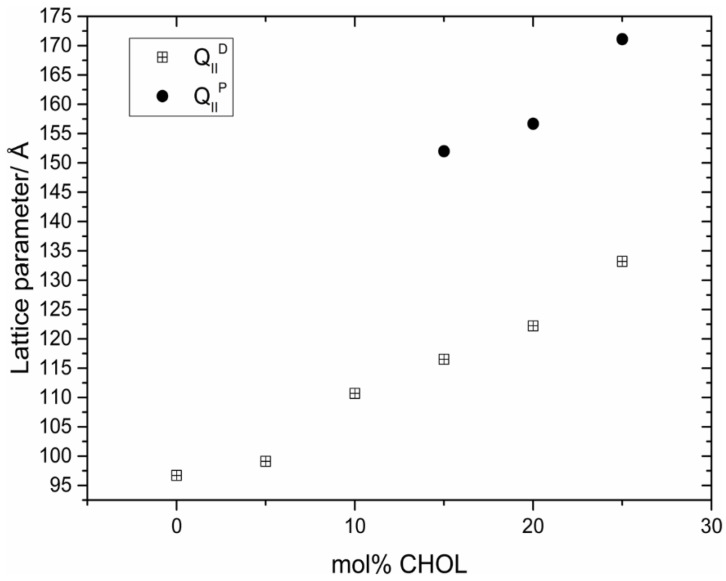
Swelling of a monoolein inverse bicontinuous cubic phase by the incorporation of cholesterol. Reproduced from [26] with permission from the Royal Society of Chemistry (UK).

**Figure 8 cells-14-00716-f008:**
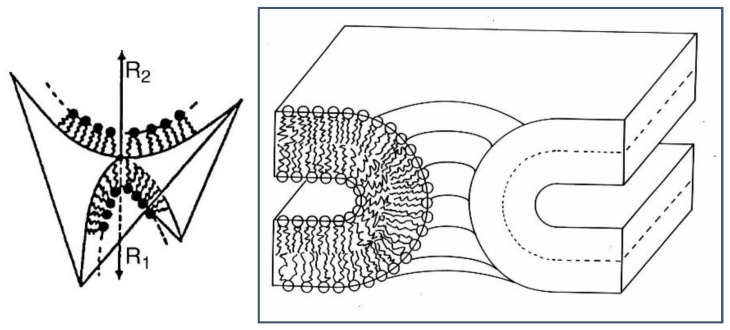
A fusion channel between two fluid bilayers is a region of negative Gaussian curvature (saddle-surface). Reprinted from [12] with permission from Elsevier.

**Figure 9 cells-14-00716-f009:**
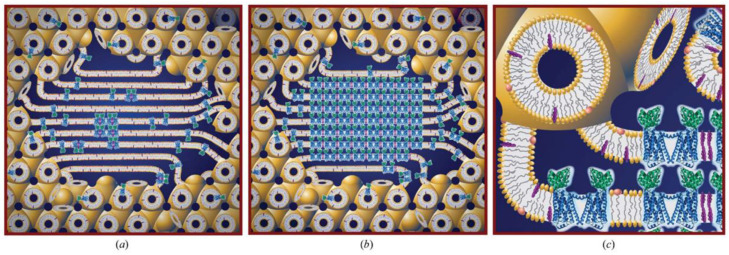
Schematic of the proposed mechanism of in cubo crystallisation of membrane proteins. Purified protein is reconstituted into the bilayer of a lipid bicontinuous cubic phase. A precipitant is added, which induces structural changes including phase separation into a coexisting lamellar phase (**a**), which acts as a conduit for protein molecules to the developing crystal (**b**). An expanded view is shown in (**c**). Reprinted with permission from [36]. Copyright 2013 American Chemical Society.

**Figure 10 cells-14-00716-f010:**
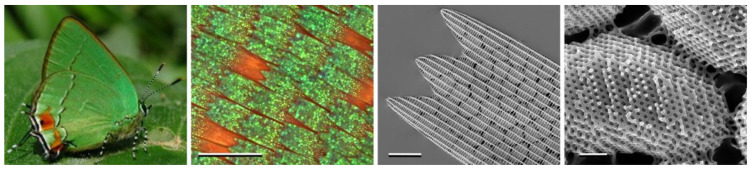
Swollen gyroid cubic crystallites in the green wing scales of the butterfly *Erora opisena* as imaged on the right by electron microscopy. Taken from [43] and used with permission of the Royal Society (UK).

**Figure 11 cells-14-00716-f011:**
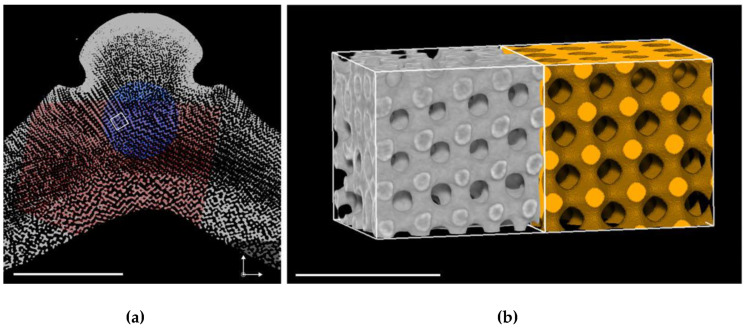
X-ray tomograph of a region of the stereom of the sea urchin *C. rugosa* that closely resembles a single Primitive periodic minimal surface. (**a**) Cross-section; Scale bar = 1 mm. (**b**) A blowup of the highlighted region (in grey) and a simulated nodal approximation of the single Primitive surface with a solid volume fraction ϕ = 0.38 (in yellow) and lattice parameter a = 30 μm. Scale bar = 100 μm. Taken from [44] and used with permission of the Royal Society (UK).

**Figure 12 cells-14-00716-f012:**
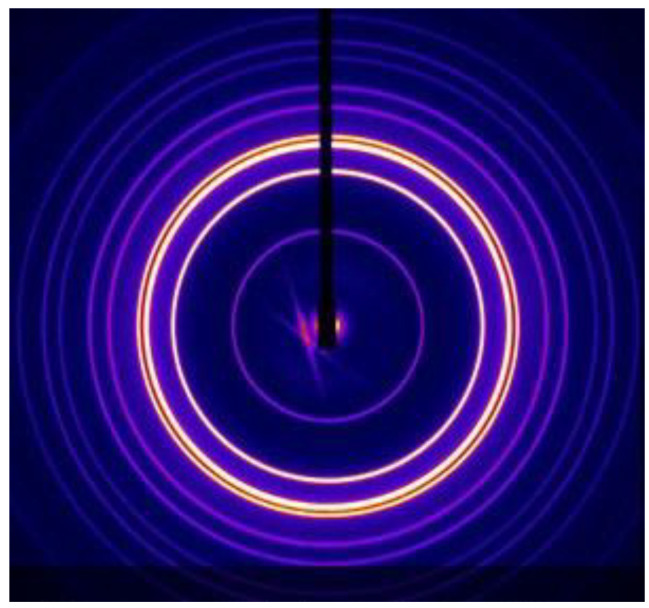
Synchrotron small-angle diffraction pattern of a fully hydrated 1:2 mixture of DOPC/DOG at 16 °C. Data are from beamline ID02 at the ESRF, Grenoble, France. Reproduced from [48] with permission from the Royal Society of Chemistry (UK).

**Figure 13 cells-14-00716-f013:**
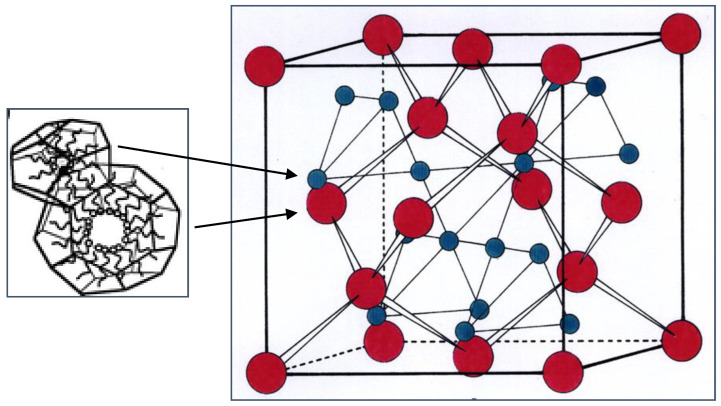
Schematic structure of the discontinuous inverse micellar cubic phase of spacegroup Fd3m. Adapted with permission from [49]. Copyright 1992 American Chemical Society.

**Figure 14 cells-14-00716-f014:**
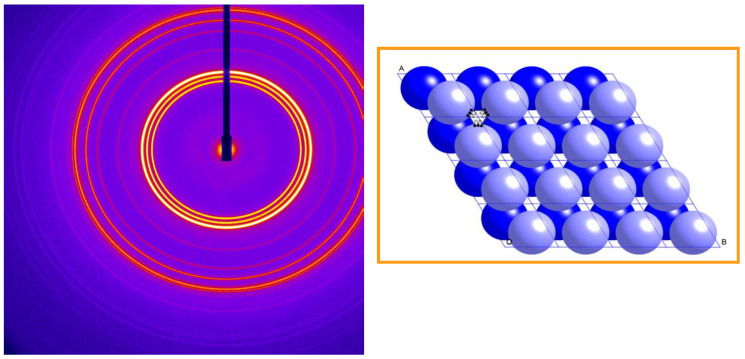
The small-angle X-ray diffraction pattern (**left**) of a fully hydrated 1:2:1 mixture of dioleoyl phosphatidylcholine, dioleoyl glycerol, and cholesterol, at room temperature. The schematic structure is also shown (**right**). The different colour shading of the two inverse micelles within the unit cell is purely for artistic effect. Adapted from [55]. Copyright 2009 American Chemical Society.

**Figure 15 cells-14-00716-f015:**
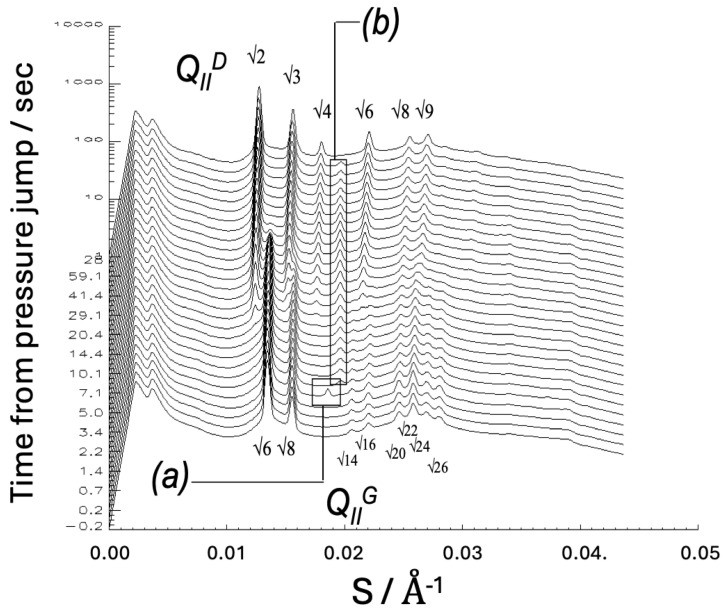
Time-resolved synchrotron X-ray diffraction measurements on monoolein, showing an Ia3d (G) cubic to Pn3m (D) cubic phase transition, induced by a pressure-jump from 600 to 240 bar at 59.5 °C. Reprinted from [62] with the permission of the American Physical Society.

**Figure 16 cells-14-00716-f016:**
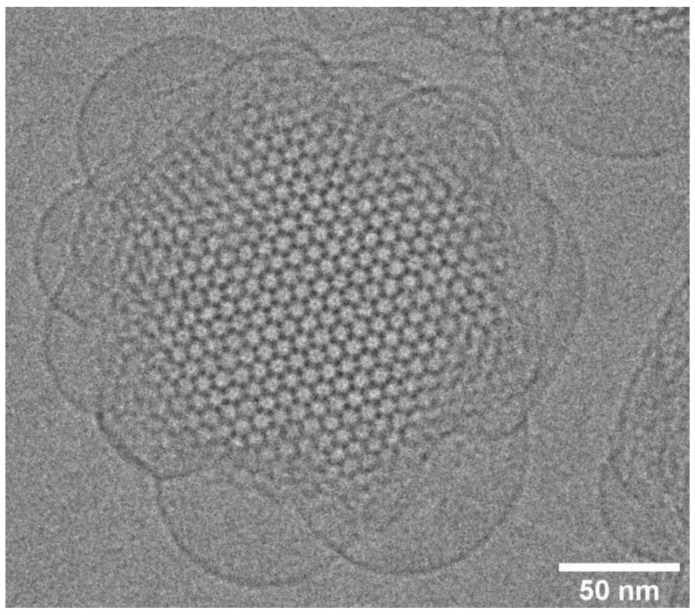
Cryo-EM image of an Im3m cubosome of monololein with 5 wt% of the block copolymer F127. The view appears to be along the [111] crystallographic direction. Image courtesy of Hanna Barriga.

**Figure 17 cells-14-00716-f017:**
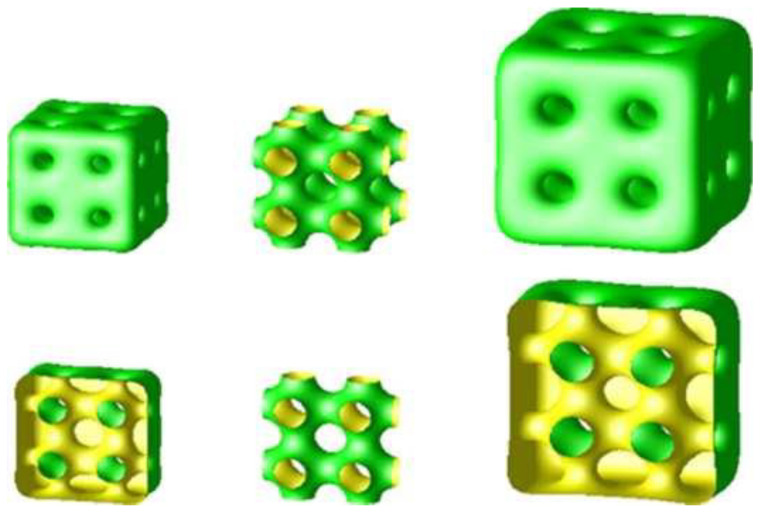
Theoretical modelling of cubosome geometry and topology, as a function of unit cell size (the cubosome on the right has 2.5 times the bilayer surface area of the one on the left). The cubosome structure shown corresponds to the bulk cubic phase of spacegroup Im3m, based on an underlying P minimal surface, shown in the centre column. The lower row shows vertical sections through the structures of the upper row. Reprinted with permission from [67]. Copyright 2015 American Chemical Society.

**Figure 18 cells-14-00716-f018:**
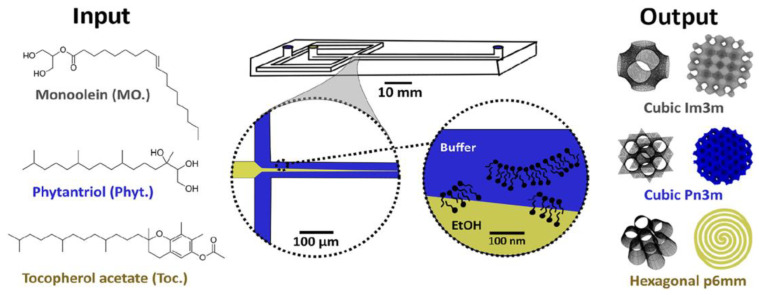
A microfluidic hydrodynamic focussing technology for the production of cubosomes and hexosomes. Reprinted from [77] and used with permission of Springer Nature.

**Figure 19 cells-14-00716-f019:**
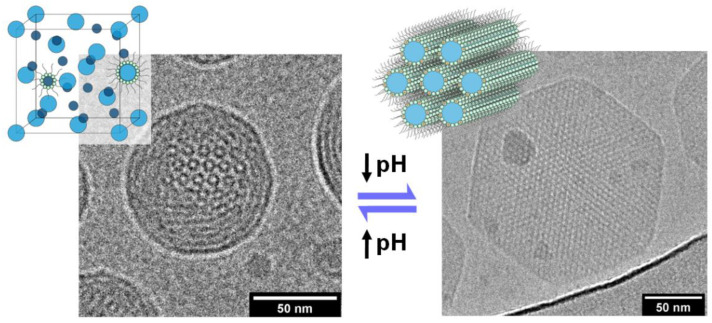
Cryo-TEM images of a pH-sensitive lipid nanoparticle which converts from an internally confined Fd3m micellosome structure (**left**) to a more porous H_II_ internal structure (**right**) upon lowering the pH below 6. The conversion is reversible upon increasing the pH back to pH 7.4, as indicated by the arrows. Reprinted with permission from [79]. Copyright 2021 American Chemical Society.

**Figure 20 cells-14-00716-f020:**
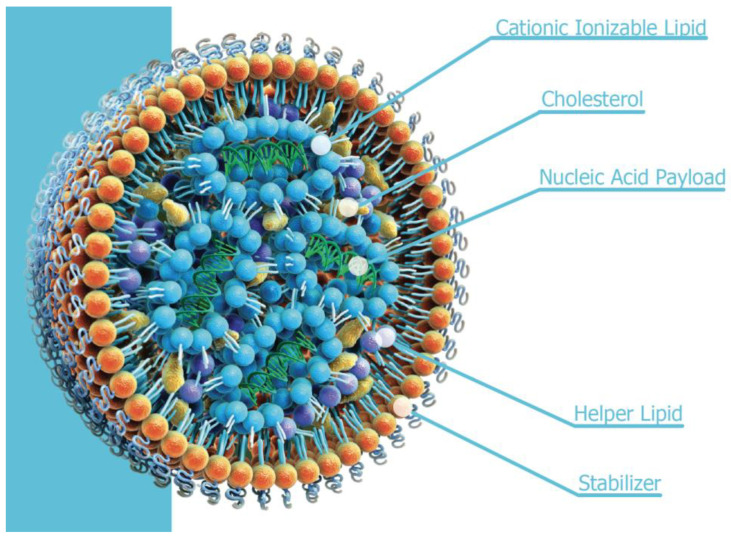
A schematic representation of a lipid nanoparticle with encapsulated nucleic acid. ©2025 Cytiva—Reproduced with Permission of Owner.

**Figure 21 cells-14-00716-f021:**
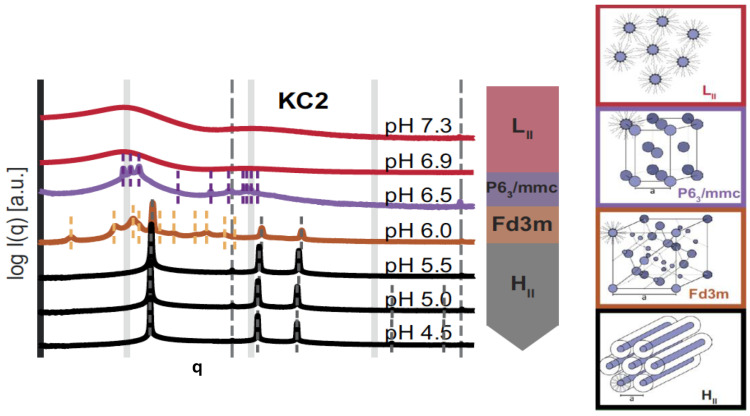
The sequence of inverse phases observed by small-angle X-ray diffraction in a cationic ionizable lipid (KC2)/cholesterol 3:1 bulk mixture upon lowering the buffer pH from 7.3 to 4.5. Lattice parameters are denoted by the letter (a) in the right-hand schematic structures. Adapted from [80] with permission from PNAS.

**Figure 22 cells-14-00716-f022:**
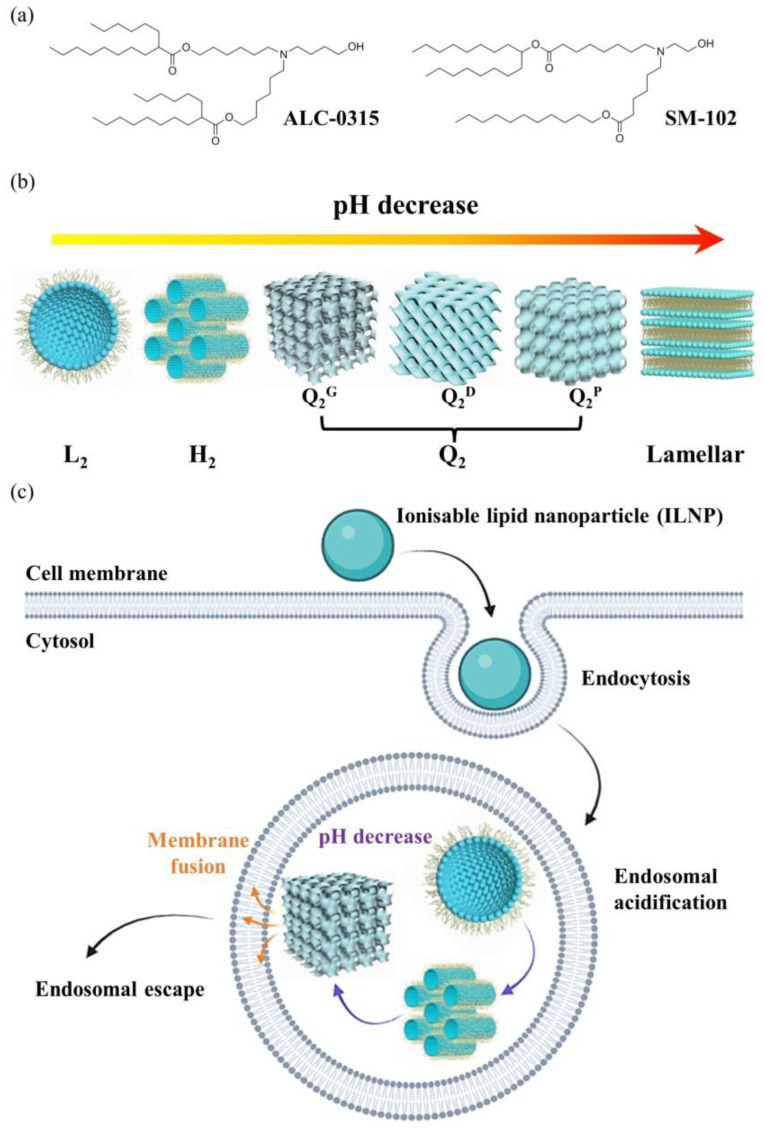
A possible mechanistic pathway for the endosomal uptake of ionizable lipid nanoparticles into cells, where endosomal acidification leads to phase transitions to inverse hexagonal and then inverse bicontinuous cubic phases, promoting nanoparticle fusion with the endosomal membrane and release of contents. (**a**) Chemical structures of ionizable lipids. (**b**) Sequence of phases upon lowering the solution pH. (**c**) Proposed pathway for uptake in cell by endocytosis, and subsequent endosomal escape. Reprinted from [82] with permission from John Wiley and Sons.

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
