# Peer review of "Inverse Bicontinuous and Discontinuous Phases of Lipids, and Membrane Curvature"

_cells, 2025, doi:10.3390/cells14100716_

Round 1
Reviewer 1 Report
Comments and Suggestions for Authors
This review article is sound and gives an exhaustive overview on the topic. I would like to recommend only only poins which could improve even more the article:
1) State in section 2. that PE and PC refers to the lipid headgroup.
2) In line 81, the authors says that PC and shingomyelin have the same polar headgroups, however they look different in figure 1 what is confusing. maybe the author could chose another figure ?
3) The resolution of figure 4 is really not good and should be improved. The axes and labels of Figure 15 are also not good and should be improved.
4) Line 285: GPCR is not defined, but should be given.
5) Subtitles of section 9, 10 and 11 should go to the next page.
6) Line 478: "theoretically-modelled" without the hyphen
7) Line 508: "in vitro" in italic
8) line 539: pKa should be defined or introduced.
Author Response
Comment 1: State in section 2. that PE and PC refers to the lipid headgroup.
Response 1: Added to line 86: ‘The polar headgroup of’.
Comment 2: In line 81, the authors says that PC and shingomyelin have the same polar headgroups, however they look different in figure 1 what is confusing. maybe the author could chose another figure ?
Response 2: Added in line 81: ‘phosphocholine’.
Comment 3: The resolution of figure 4 is really not good and should be improved. The axes and labels of Figure 15 are also not good and should be improved.
Response 3: Figure 4 (p6) has been improved in resolution. Figure 15 (p13) axes labels have been improved.
Comment 4: Line 285: GPCR is not defined, but should be given.
Response 4: Added to lines 283-284: ‘(G protein-coupled receptor)’.
Comment 5: Subtitles of section 9, 10 and 11 should go to the next page.
Response 5: Done in each case.
Comment 6: Line 478: "theoretically-modelled" without the hyphen
Response 6: Change made in line 466.
Comment 7: Line 508: "in vitro" in italic
Response 7: Changed to italics in line 496.
Comment 8: line 539: pKa should be defined or introduced.
Response 8: Added in lines 528-529: ‘(the pH at which the ionizable group is half-dissociated)’.
Reviewer 2 Report
Comments and Suggestions for Authors
The manuscript is written in a clear, logical, and understandable manner, particularly for readers with a background in biophysics, physical chemistry, or cell biology related to membranes. The author defines key concepts such as interfacial mean curvature (H) and Gaussian curvature (K) in (Section 3, Figure 2) and explains the relationship between lipid molecular shape and preferred curvature (Figure 3). The progression from fundamental concepts of lipid self-assembly and phase diagrams to specific examples of inverse phases (bicontinuous and discontinuous) and their applications is well-structured. Figures are clear and support the text well. The author explicitly mentions that this review is not intended as a comprehensive review of the entire field of membrane curvature (lines 20-21) or non-lamellar phases (lines 46-47), but rather focuses on specific aspects, draw from work done in the author's group (lines 44-46), presented in the context of a plenary lecture (lines 18-19). Given this framing, the review provides sufficient information on the selected topics: lipid self-assembly principles, definitions and types of curvature, common inverse lyotropic phases (HII, bicontinuous cubic Ia3d, Im3m, Pn3m; discontinuous cubic Fd3m, Fm3m; hexagonal P63/mmc), factors influencing phase behavior (temperature, pressure, composition, pH), and selected biological and technological relevance (membrane fusion, protein crystallization, biological morphologies, lipid nanoparticles like cubosomes, hexosomes, micellosomes, and their role in drug/gene delivery). Key experimental techniques used in the field (SAXS, cryo-EM, pressure-jump studies, microfluidics) are mentioned in the context of the results discussed. The author acknowledges the vastness of the membrane curvature field and cites relevant reviews [1, 2, 4-7]. Throughout the text, key concepts and findings are attributed to foundational and contemporary studies via numerous citations, situating the discussed topics within the broader history and current state of the field (e.g., citing Luzzati, Seddon, Templer, Caffrey, Drummond, Leal, Raedler, etc.).
The main conclusions woven throughout the review are that: (i) lipids self-assemble into a variety of non-lamellar phases characterized by specific interfacial curvatures; (ii) this phase behavior can be rationally tuned by thermodynamic parameters and molecular composition; (iii) these non-lamellar structures, particularly inverse phases, play roles in biological processes and have significant potential in biotechnology (especially LNPs). The link drawn between pH-induced phase transitions in ionizable lipid/cholesterol mixtures and proposed LNP endosomal escape mechanisms (Section 13) is a pertinent example where the presented results directly underpin the discussed conclusions.
The manuscript is already a strong and informative review within its stated scope. A minor suggestion to potentially enhance it further would be:
- The author could consider adding a brief concluding section. While the review flows logically, it ends somewhat abruptly after discussing the proposed LNP escape mechanism (Figure 22). A short summary could reiterate the main themes – the prevalence and significance of inverse lipid phases, the importance of interfacial curvature as a concept, the tunability of these systems, and their diverse roles in biology and nanotechnology. It could also perhaps offer a very brief perspective on remaining challenges or future directions in the study and application of these fascinating structures.
Author Response
Comment 1: The author could consider adding a brief concluding section. While the review flows logically, it ends somewhat abruptly after discussing the proposed LNP escape mechanism (Figure 22). A short summary could reiterate the main themes – the prevalence and significance of inverse lipid phases, the importance of interfacial curvature as a concept, the tunability of these systems, and their diverse roles in biology and nanotechnology. It could also perhaps offer a very brief perspective on remaining challenges or future directions in the study and application of these fascinating structures.
Response 1: The following paragraph has been added in page 20:
‘In summary, this review article has described the self-assembly of phospholipids in aqueous solution into a rich variety of lyotropic liquid-crystalline phases. I have focussed attention on inverse nonlamellar phases, having negative interfacial mean curvature. I have discussed how the phase behaviour can be controlled, and the structures of the phases tuned, either by varying thermodynamic parameters such as temperature, hydrostatic pressure, hydration, or solution pH, or by adding various amphiphilic molecules such as diacylgycerols, cholesterol, or ionizable lipids. I have shown how such nonlamellar phases are intimately related to dynamic processes involving topological transformations in membranes, such as membrane fusion. Furthermore, there is now compelling evidence that nonlamellar phases are utilized by Nature in certain specialized structures in cells, for example in templating the structures which exhibit structural colour in the wing cells of certain butterflies. There is much still to be discovered about the role of such complex self-assembled lipid structures in Nature. Nonlamellar lipid phases, when dispersed into lipid nanoparticles, also very attractive candidates as vehicles for the delivery of drugs and nucleic acids into cells. We can expect to see the highly-successful approach taken for Covid vaccines to be extended to many other areas of molecular medicine in the near future.’